# The Relationship between Neuromuscular Control and Physical Activity in the Formation of the Visual-Psychomotor Schemes in Preschools

**DOI:** 10.3390/s21010224

**Published:** 2020-12-31

**Authors:** Roxana Buzescu, Florentina Nechita, Silviu Gabriel Cioroiu

**Affiliations:** 1The School for Moving Health, SANpitic from Brasov, 500035 Brasov, Romania; roxana.buzescu@gmail.com; 2Department of Physical Education and Special Motricity, Faculty of Physical Education and Mountain Sports, Transilvania University of Brasov, 500068 Brasov, Romania; silviu.cioroiu@unitbv.ro

**Keywords:** psychomotor development, neuromuscular control, physical education, preschoolers, health

## Abstract

*Background:* This research has started from the empirical observation that preschoolers who practice systematic and continuous physical activities can solve the tasks they receive more accurately and in less time than those who do not do sports in an organized setting. *Methods:* The research was carried out in 2015 in the Laboratory of Physical Therapy and Special Motricity of the Faculty of Physical Education and Mountain Sports, Transilvania University of Brasov. The survey sample included 51 preschoolers (26 boys and 25 girls), and the study implemented “real experiment” type research with a post-test phase to find out to what extent cortical stability is dependent on practicing a form of systematic movement at the ages of 4–6 years by analyzing proprioceptive sense and neuromuscular control. Thus, we could see how a 4-to-6-year-old child’s brain responds to a given stimulus by using the ERGOSIM condition simulator, which provides real-time feedback. *Results:* The results of the study show significant values for the visual control of the subjects by adjusting movement. *Conclusions:* The practice of physical activities benefits from learning through the visual scheme, having real-time feedback, and subjects being able to maintain indices closer to the required model, on the one hand, and on the other, to return with spherical correction stimuli during a wrong move much better than those in the control group. The results suggest that systematic practice of psychomotricity can improve general development and cognition in children, and that implementing this methodology could thus be useful in educative intervention.

## 1. Introduction

The objectives of this research started from the following questions, which were born from empirical observations made during work with children aged 4 to 6 years:

1. Are there differences in the formation of a psychomotor scheme in children aged 4–6 years who participate in physical exercises systematically and continuously compared to those who participate occasionally? How fast can a child aged 4 to 6 years manage to change movement parameters according to visual stimulus?

2. How long can children keep their attention focused after learning how much force to apply in motion and how much amplitude to give it? Are there differences in maintaining attention between those who exercise systematically and those who do so occasionally or not at all?

3. Does age and, therefore, the degree of psychomotor development influence neuromuscular control and the speed of adaptation of the motor act depending on the visual stimulus?

4. Is the conscious and concentrated performance of a motor act superior to a motor act that has become reflexive?

From these objectives, the following research hypotheses were outlined:

-The first hypothesis is to discover to what extent cortical stability is dependent, at the age of 4–6 years, on the practice of a systematic form of movement, by analyzing the proprioceptive sense and neuromuscular control.

-The psychomotor learning process, with real-time feedback, is superior to the time and quality of information processing in children aged 4 to 6 years who practice systematic exercise compared to those who do so occasionally.

-Maintaining attention in a system of information retrieval, i.e., learning, is different in children aged 4 to 6 years who practice systematic exercise compared to those who do it occasionally.

Psychomotor intervention is an activity that is performed in order to enhance an individual’s potential development through the use of the body, in action and in motion [1].

This is a complex function that integrates and unites the motor and mental elements determining the regulation of the individual’s behavior. It includes the participation of various mental processes, thus ensuring adequate performance of response to different situations/stimuli [2], as well as adequate psychomotricity [3]. Being a basic function with processes and phenomena of mental nature, psychomotor intervention is generated and expressed by involuntary movements of the body, influencing their application in actions [4].

Psychomotricity theories play an important role in the development of preschool children and are necessary for the educational process at this age, highlighting the possible combinations of conditioning and coordination of skills that characterize the motor-coordination component of human kinesthetic potential [5].

The selection of the most effective methods of physical activity defined not only by their contribution to the education of the main components of psychomotricity, but also by their structure and their degree of accessibility for the ages being researched represents a key step in the didactic strategy for the development of psychomotricity in children aged 4–6 years [6].

Psychomotor education is the beginning of the process of early childhood education. The learning disabilities detected in a child may cause psychomotor development delay [7].

In psychomotor activity, the movements and basic elements of gross and fine motricity are formed. Psychomotor education aims for the optimal development of psychomotricity components in children through the organized use of means, materials and methods corresponding to children’s particularities and their level of training. Thus, the ability of the locomotor apparatus to manipulate various objects or to perform certain movements in relation to space and time relations bears the name of coordination [8].

Coordination–neuromuscular control is the ability of the human brain to engage in voluntary activities that engage one or more skeletal muscles. Optimal neuromuscular control requires increased involvement and attention, as well as awareness of the movements that are required to be performed [9]. Thus, the development of visual perception is achieved by identifying the presence and the degree of visual–perceptual deficit and visual integration in children aged between 4 and 12 [10].

The effects that occur in this process are:-The child becomes aware of the gestural space through various sedentary positions, executed in diverse postures (horizontal and vertical positions of the arms in the straight position, the position of the legs);-The child gets to know his body through manual contact (a priceless source of exteroceptive sensations) and through personal actions on their own body; the individualization of the body through manual contact personal actions on the child’s own body, with left–right discrimination, executed by a child who has already gone through and passed the previous stages [11].

The essence of the research is to use the “hand–eye coordination” component of motor–eye coordination, which is defined as the ability of the vision system to analyze and guide the information captured by the eyes to the upper limbs that are tasked with carrying out ordered actions, such as catching or throwing an object.

Recent studies have confirmed that hand–eye coordination is based on the ability of eye perception to aid the performance of a task by the hand [12,13,14,15,16,17]. For this to happen, the ability to perceive details, good control of the hand–eye coordination process and unaffected motor independence of the eyeball are necessary, because the hand movements must be as effective as possible. Weakness, instability, muscle tension or hypertonia are also causes that cannot be overlooked, as thanks to them the fine movements are transformed into interrupted or incorrectly performed movements along the amplitude trajectory [18].

Ages of 4 to 6 can be a degeneration factor in the neuromuscular coordination process, but physical activity performed with moderation tends to use both gross and fine motor skills and can successfully combat the effects of possible delays, aiming at optimal development of the neuromuscular system [19]. Controlling the action of the body and the hands is possible due to the level of intelligence, managing to give rise to the coordination process. This intelligence time concerns the actions performed by both the gross and the fine movements in which the body engages [20].

Cognitive therapy is the modification of the cognitive schemes of the subject [21]. The cognitive system’s stimulus processing is made from physical characteristics (contour, color, dimensions, displacement, etc.) and semantic or functional characteristics (meaning, function, category to which it belongs, etc.). This characteristic of the cognitive system is called the ascending analysis of the stimulus. It contains both the properties of peripheral cognitive modules and the characteristics of the stimulus. The processing of this system, from the subject’s knowledge to the physical properties of the stimulus, is called downstream analysis.

The computational level refers to the processing required for input-to-output transformations or the input–output function. It shows the exhaustive determination of the processes to which the data of the problem (input) for obtaining the solution (output) are subjected.

The primary processing of visual information is based on computational theory, which has an abstract, formal–mathematical character, and which logically and mathematically reflects the function by which a cognitive system makes an input to correspond to a specific output. This theory explains the reception of contours and three-dimensionality. Primary processing involves prenatal processing that represents the physical characteristics of the stimulus in the cognitive system. Processing at this level localizes the stimulus but does not characterize it.

Secondary visual perception contains mechanisms for recognizing figures and objects (their result is three-dimensionality and stimulus identification). Color processing is based on chemical and physical phenomena and mechanisms. Secondary processing in visual information (object recognition) has as input the intermediate sketch, and as output three-dimensional representations.

Recognition is the superposition of perceived object over its representation in memory; it is characterized by speed and flexibility. An important role in this is played by ascending analysis, but also downstream analysis. The relationship between intermediate representations and representations of objects is facilitated by spatial details or “non-incidental properties”: the straight line, the curve, parallelism and symmetry.

The novelty of the study is to correlate the level of neuromuscular control with physical activity in the formation of visual–psychomotor schemes at this age.

The purpose is to determine whether the degree of psychomotor development with real-time feedback is superior in terms of the time and quality of information processing as an adaptation of the motor act according to the visual stimulus.

## 2. Materials and Methods

### 2.1. Study Lot

The participants were children aged between 4 and 6 years from private kindergartens in Brasov with teaching and activities in German and English at ”Heidi Kindergarten and Maya and Montessori Kindergarten”. The children came from a medium-to-high socio-economic environment, representing a homogeneous sample in terms of quality of life.

The research includes a cohort of 51 (26 boys and 25 girls) preschool children divided into two groups: The experimental group (LE) comprised 29 subjects who practiced systematized and continuous physical exercise for at least 1 year at least 2 times a week, and the other group 22 (the control group—LC) who practiced organized physical activities sporadically or not at all.

The sample consisted of: LE: 18 (62.07%) females and 11 (37.93%) males; LC: 7 (31.82%) females and 15 (68.18%) males. The average age was 5.06; the minimum age was 4.2 and the maximum age was 6.02.

The research was based on informed consent, which aims to inform parents about the research and includes the purpose, objectives, methods and procedures used, and there were no negative incidents during the experiment.

### 2.2. Procedure

In order to observe how the 4-to-6-year-old child’s brain responds to a given stimulus, an ERGOSIM condition simulator that offers real-time feedback was used. ERGOSIM is a highly successful program based on a computer-assisted simulator. It was designed by the National Research Institute for Sport, Bucharest, Romania.

The novelty and originality of the device, which also gives it its uniqueness value, is its ability to analyze the motion parameters at work at the desired speed on an acceleration motion achievable in all planes at a multitude of angles with ideal amplitude, no forced return, as happens with classical devices (arc, chain, ribbon, etc.).

The realization of the psychomotor–visual schemes included the following procedures on ERGOSIM:

a. Test 1 (T1): Non-visual execution, Figure 1.

Initial position: seated at the same time as holding the cane with two hands, grasping from above at shoulder width, arms outstretched. Performing 10 tractions (lowering the cane forward to the abdomen).

Generally, cortical stability followed the same route without any explanation. For this test, only the lines coming out of the pattern, the eccentric ones, were counted, which led to the percentage of cortical instability.

b. Test 2 (T2): The execution with visual stimulus in imitation of the model rendered by the simulator, Figure 2.

When testing T2, the difference from T1 was that the subjects received the information that the executed movement must draw a line as close to the model (Figure 3—yellow graph) 10 times. They did not receive any explanation as to how the movement should be done (how hard or slow to pull or how to reach their arms).

Thus, subjects achieved 20 tractions, helmet type, assisted by ERGOSIM, on a given model. From the number of executions, 10 of these were executed without looking at the screen, being the first part of the T1 test, and in the next 10 (T2), the children tried to draw a drawing on the screen as close as possible to the given pattern. For each traction on the screen, a graph appears based on the amplitude, velocity, and force of the movement. For each movement on the screen, a line of a certain color appears—see Figure 3. The first strokes were set to appear on the monitor in green, then gradually to red until the end.

The first 5 tractions are colored with green shades from closed to open and the next 5 from orange to red very dark, so they can be evaluated individually from a bundle of 10.

When testing T1 (Figure 3), we can see 3 “atypical” blows, which are offset from the order chart for the other tractions. We notice that all the other 7 tractions have a fast ascending route of force (up to about 0.05–0.06 m), and the 3 tractions highlighted do not fit in the pattern of most orders. The one marked by the blue arrow has a slow ascending path; a force of 2dN is recorded at a position of 0.1 m. The ascent is long delayed compared to the rest of the tractions.

The traction highlighted by the yellow arrow also bends because it reaches 2 dN very late, i.e., at about 0.25 m, compared to most of the tractions, which reach this threshold of the model before 0.07 m.

The traction marked with the white arrow does not reach the value of the model and does not have a fast-ascending and slow-descending path. The appearance is closer to some force–position function.

Thus, the graph of proprioceptive sense with eyes open but without real-time feedback, as noted with note 7, is appreciated; the three strokes described above that reduce from the general pattern of beginnings for the others are subtracted from the total of 10.

The T2 test noted at which traction subjects managed to approach the model (the first traction they learn). Afterwards, until as many tractions as possible, subjects were concentrated on trying to follow the pattern (the first traction they look at). The total number of successful tractions (the note obtained by seeing the model) and maintaining the focus of attention (the number of the tractions maintain learning) was calculated by the difference between “the first traction they look at” and “the first traction they learn”.

We appreciate the number of reactions that the subject’s brain has been able to transmit commands to move the motion parameters to the given pattern.

Thus, for example, the green line, shown in Figure 4 with a yellow arrow, is the second or third traction in which the subject manages to send the command to the effectors. It is important to know that the brake on which the subject pulls was set to 0, because we did not want to test the strength of the children, but only their ability of neuromuscular control. It is knowns that using a large brake, neuromuscular control is easier to achieve, and the model is easier to reach; thus, any movement can be performed much more easily.

The ideal of motricity is to perform different movements with the same amplitude and direction on a predetermined volume and intensity with minimal energy consumption.

Another example, in the graph above, is the red coral line, shown by the purple arrow (one of the last tractions). This was counted as good traction, because up to 15 cm, the subject managed to stay on the model, then between 16 and 30 cm he could not control it, but from 33 until the end, he again tried to reach the model (Figure 4).

### 2.3. Data Analysis

The results were processed with IBM SPPS Statistics 20 (Armonk, NY, USA). The statistical indicators used were arithmetic average (X), standard deviation (SD) and Student’s *t*-test (*t*), for *p* < 0.05.

## 3. Results

This section presents the most relevant results and descriptive information of the study on the procedure for the realization of the visual–psychomotor schemes by the preschoolers included in our study.

Following statistical processing, the values of the statistical indicators presented in Table 1 were extracted from the SPSS outputs.

According to our study using the ERGOSIM simulator, the most important index was “visual stimulus note”, which has a value of *t* = 6.61 and a significance threshold of *p* <0.01. There were significant differences when participants had visual control and could use visual input to adjust their movement.

Another significant difference (Table 1) is “the number of tractions in which learning is maintained”, at which the difference was 2.67 with *t* = 4.33 at *p* < 0.01, which shows that practicing systematic and continuous sports helps in concentrating attention over a longer period of time or returning to the task of learning based on previous learning.

In terms of “first learning traction” and “first shot look”, differences are 1.08 and 1.59 with *t* = 1.79 and *t* = 1.92, respectively, for *p* < 0.5. Thus, practicing physical exercise in preschool children does not bring improvements to the requirements and processing of information until the problem is solved (learning), as well as when the task is performed over a longer period of time.

There are no notable gender differences for the values with high significance, so for the number of blows with visual stimulus, the boys had an average of 4.42 and the girls 4.52. The same is true for the number of the tractions, where the average value for boys was maintained at 3.27, and for girls 3.32. The advantage that girls have is imperceptible and may be due only to the seriousness of the approach to that activity.

Regarding differences according to age, and neuromuscular control increasing as biological age increases, we did not find a higher share of it with growth. We have grades of 7 and 8 at the age of 4.2 years (with visual stimulus) which we do not find at all at 6 years, and for the number of tractions, at the average age of 5.06 years, 5 children had an average of 5.20 tractions (the maximum age being 6.02 years). Thus, neither gender difference nor age differences influence the processing of information until the problem is solved and the task is performed.

## 4. Discussion

Through our results, our study helped to discover to what extent cortical stability is dependent, at the ages of 4–6 years, on the practice of a systematic form of movement by analyzing proprioceptive sense and neuromuscular control. The results of the study are in correlation with previous studies. The study highlighted the correlation between cortical stability (defined by strength, balance and attention–concentration) and individuals who exercise and those who do not, at the age of 4–6 years.

Although many studies have highlighted the importance of “hand–eye” research, the computational mechanisms underpinning these coordinated movements remain elusive [22,23].

The results of these studies show that sustained exercise contributes to increased performance [24] when visual cues are allowed, which is closer to real-life situations in which children can use their eyesight.

The possibility of evaluation and real-time observation of behavior allows an optimal intervention in the direction of psychological aspects [25].

Based on the importance of practicing physical activities systematically and continuously [26], the focus of the attention has increased over a longer period of time. From the point of view of the number of tractions maintained, our research shows that preschoolers return to the execution of the task imposed on the basis of previous learning (LE). In the case of LC, physical exercise does not bring improvements to the requirements and the processing of information until the problem is solved (learning), as well as when carrying out the task for a longer period of time.

A study aimed at assessing knowledge about visual motor integration determined that this is a vital ability in childhood development, which is associated with the performance of many functional skills, and this research confirms our aspects and results [27].

After all, the analysis of the research is the consequence of stabilizing the relationship between physical activity and the neuro-motor coordination of the subject. It is certainly more important to place emphasis on movement at this age, whether practiced under the supervision of a non-specialized framework in the field of physical education and sports, or the exaggerated strain in modern learning algorithms [28]. Cognitive learning [29,30], with this neurobiological base formed, will have a solid foundation in our study.

Limitations: In this paper, there are still some limitations:

Our study employed a small number of subjects, and expanding research could target pre-university cycles.

Cognitive factors contribute significantly to loss of attention, so children at the age of 4–6 years master many physical abilities that must be well correlated with specific activities performed in their own environments.

Strengths: The strengths of the study were the evaluation of the two groups through the complex ERGOSIM [31,32] conditions simulator at this age and the relevance of the results obtained and the findings of the study.

## 5. Conclusions

The study highlighted the idea that preschoolers, when they do not have a visual schema as required learning information, can maintain effective neuromuscular control over movements, developing similar strength–function position reports between successive repetitions. The application of research within the educational process has highlighted the importance of practicing systematic and continuous physical activities.

The results of the research allow referrals to the fact that based on visual schemes, by internal representations of physical and mental actions, these children in the experimental group evolved through assimilation and accommodation to superior neuromuscular control in order to approach the given model.

## Figures and Tables

**Figure 1 sensors-21-00224-f001:**
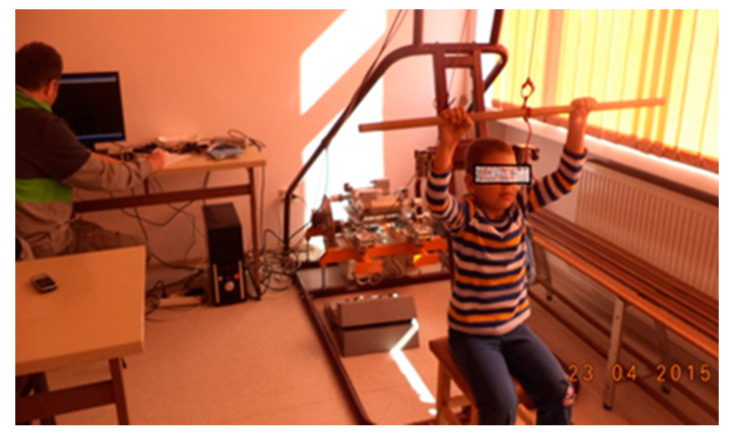
Non-visual execution, ERGOSIM.

**Figure 2 sensors-21-00224-f002:**
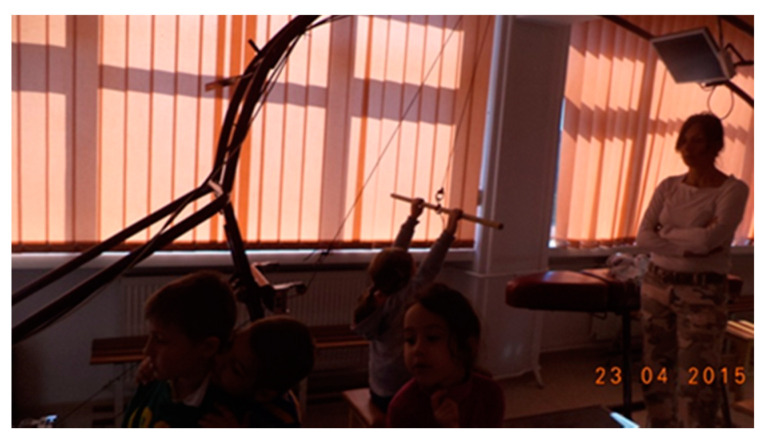
Execution with visual stimulus, ERGOSIM.

**Figure 3 sensors-21-00224-f003:**
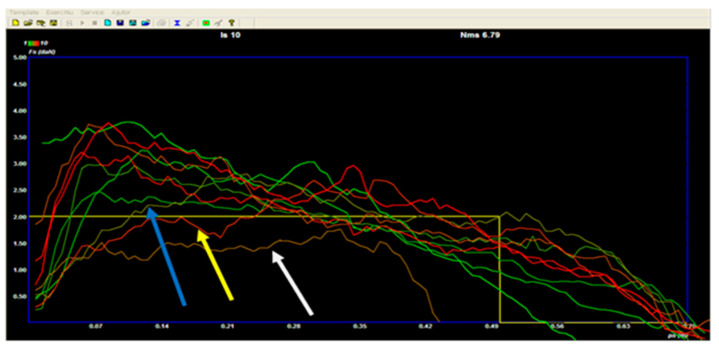
The visual–psychomotor scheme made without visual stimulus, Subject L. M.

**Figure 4 sensors-21-00224-f004:**
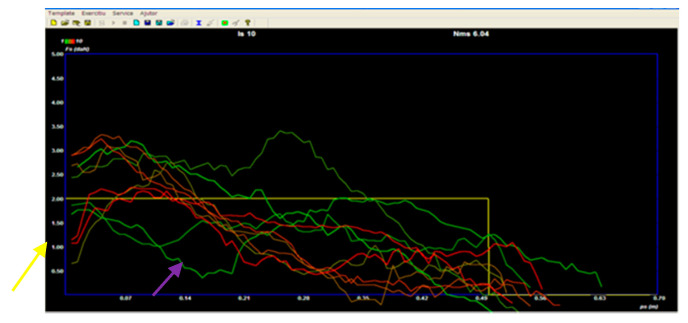
Visual–psychomotor scheme with real-time model tracking, Subject L.M.

**Table 1 sensors-21-00224-t001:** Statistical description of the research activity indexes.

Group		No.	X	SD	Student Test t *p*
Note without visual stimulus	LE	29	6.86	0.86	1.89 0.06
	LC	22	6.00		
The first traction he learns	LE	29	2.69	1.08	−1.79 0.08
	LC	22	3.77		
The first traction he looks	LE	29	7.14	1.59	1.92 0.06
	LC	22	5.55		
Visual stimulus note	LE	29	5.97	3.46	6.61 0.01
	LC	22	2.50		
The number of the	LE	29	4.45	2.67	4.33 0.01
tractions maintain	LC	22	1.77		

No.—number of subjects; X—Arithmetic average; SD—standard deviation; *t*-test; *p* < 0.05; LE—The experimental group; LC—The control group.

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
