# Peer review of "The Relationship between Neuromuscular Control and Physical Activity in the Formation of the Visual-Psychomotor Schemes in Preschools"

_sensors, 2020, doi:10.3390/s21010224_

Round 1
Reviewer 1 Report
First of all, I would like to acknowledge the opportunity to review the manuscript entitled “The relationship between neuromuscular control and physical activity in the formation of teh visual - psychomotor schemes in preschools”. This shows an emerging and interesting topic in the field of education and health, in addition, the article follows the line of the journal.
In my opinion, I think the manuscript can be accepted, although some important changes are recommended:
Abstract: It is consistent, well structured and details the most relevant aspects of the manuscript.
Introduction: It addresses the topics of interest associated with research, with special relation to psychomotor development. Some recommendations would be:
- This work could help improve the rationale for this section: http://www.journalshr.com/index.php/issues/79-vol-12-suplemento-2020/395-85-llorente-cantarero-fj-gil -lozano-p-2020-effect-of-exercising-freely-during-the-school-break-on-the-cardiorespiratory-fitness-in-children-journal-of-sport-and-health-research-12supl-185 -96
- It is recommended to include a research question
- Since an experimental study is carried out, it is recommended to include study hypotheses.
Material and method:
The description of the procedure and instruments is appropriate and correct for an investigation of this nature. However, authors must:
- Specify participant selection method: convenience, randomized, etc.
- Indicate% according to gender, average age and minimum and maximum age.
- Detail criteria for experimental group and control group.
- Detail differences in the program for the experimental group and the control group.
- Indicate if the study obtained the approval of the ethics committee of your university.
- The process focuses on the technical aspects of the system, but the process followed to obtain permits for human experimentation (informed consent of parents, ethics committee, etc.) is not sufficiently explained.
- Detail if there were experimental deaths.
- The data analysis section should be developed further. What statistical tests were performed? (T-test of related samples)
Results
The statistical test is correct. However, two actions are required:
- Determine normality of the data.
- Calculate the effect size and the confidence interval for the variables in the differences between groups (this is mandatory).
Discussion
- It presents important limitations. The first is that it is very short and the results are hardly discussed. Please, the authors should delve deeper into their findings and contrast them with those of other current works. They have only included 2 quotes in the body of the discussion, and this needs to be expanded considerably.
- Data from statistical tests should not appear in the discussion. This is specific to the results (authors must move this information to the results)
- Your study has more limitations, please, you should develop them.
References
- Review according to the journal's regulations.
Reviewer 2 Report
it is my honor to review this paper, the below is my considerations.
line 34-35: delete this sentence. not saying aim of study at the beginning of the introduction section.
the structure of the introduction section should be well organized, please revise it greatly.
how did you get a sample of 51? please specify it.
how did you allocate sample to control and intervention groups?
more details of statistical analysis should be reported intead of a simple sentence.
is the study having a set of result? (only table 1)
like the structure of introduction, the organization of discussion section must be re-constructed. it is of great importance.
the conclusion should be concise.
Round 2
Reviewer 1 Report
Authors have considered most of the suggestions appropriately.
Reviewer 2 Report
the authors have addressed my concerns